# AhR Activation Ameliorates Intestinal Barrier Damage in Immunostressed Piglets by Regulating Intestinal Flora and Its Metabolism

**DOI:** 10.3390/ani14050794

**Published:** 2024-03-04

**Authors:** Xiaomei Wu, Yalei Zhang, Mengyao Ji, Wen Yang, Tanjie Deng, Guanyu Hou, Liguang Shi, Wenjuan Xun

**Affiliations:** 1School of Tropical Agriculture and Forestry, Hainan University, Haikou 570228, China; xiaomeiwu0929@163.com (X.W.); zhongsh8114@163.com (Y.Z.); changye970909@163.com (M.J.); onlyoneyangwen@outlook.com (W.Y.); dtj1302604492@163.com (T.D.); 2Tropical Crops Genetic Resources Institute, Chinese Academy of Tropical Agricultural Sciences, Haikou 571100, China; guanyuhou@126.com (G.H.); shiliguang123@126.com (L.S.)

**Keywords:** aryl hydrocarbon receptor, 6-formylindolo (3,2-b) carbazole, Cardamonin, piglets, intestinal barrier, short-chain fatty acid, metabolism

## Abstract

**Simple Summary:**

Weaning stress is a major problem in the pig industry, causing high rates of diarrhea and poor performance in piglets. Cardamonin is a bioactive plant extract that possesses anti-inflammatory, anti-oxidant, and anti-viral properties. In this experiment, we investigated the effect of Cardamonin on intestinal barrier damage and inflammatory response caused by immune stress in piglets by establishing a stress model of LPS. Compared with those of the LPS group, the intestinal mucosal morphology and expression of tight junction proteins in the cardamonin group were improved, along with the inflammation response. Furthermore, certain intestinal microbiota and inflammation-related metabolites showed a significant correlation. The results indicate that the administration of Cardamonin can effectively improve intestinal mucosal barrier damage and inflammatory response induced by LPS by regulating the composition and metabolism of intestinal microbiota, with a better effect observed at a dosage of 6 mg/kg.

**Abstract:**

The primary factor leading to elevated rates of diarrhea and decreased performance in piglets is immunological stress. The regulation of immune stress through the intestinal flora is a crucial mechanism to consider. In total, 30 weaned piglets were randomly allocated to five groups: the basal diet group (Control), basal diet + lipopolysaccharides group (LPS), basal diet + 250 μg/kg 6-Formylindolo [3,2-b] carbazole + LPS group (FICZ), basal diet + 3mg/kg Cardamonin + LPS group (LCDN), and basal diet + 6mg/kg Cardamonin + LPS group (HCDN/CDN). The results showed that compared with those of the LPS group, the expression of tight junction proteins (occludin; claudin-1) in the FICZ group was significantly increased, and the mRNA levels of IL-1β and TNF-α were significantly reduced (*p* < 0.05). HCDN treatment had a better effect on LPS-induced intestinal barrier damage in this group than it did in the LCDN group. HCDN treatment leads to a higher villus height (VH), a higher ratio of villi height to crypt depth (V/C), higher tight junction proteins (ZO-1; occludin), and higher short-chain fatty acids (SCFAs). In addition, correlation analyses showed that *Succinivibrio* was positively correlated with several SCFAs and negatively correlated with prostaglandin-related derivatives in the FICZ group and CDN group (*p* < 0.05). In summary, Cardamonin alleviates intestinal mucosal barrier damage and inflammatory responses by regulating the intestinal microbiota and its metabolism.

## 1. Introduction

The widespread adoption of early weaning techniques for piglets has become prevalent in modern intensive farming to enhance sow productivity, increase the annual number of piglets produced, and improve equipment utilization rates for greater economic gains. However, during the weaning period, piglets are susceptible to multiple stressors (nutritional, physiological, and environmental), resulting in reactions such as loss of appetite, indigestion, slow growth, diarrhea, and even death, which cause serious economic losses in the pig farming industry [1,2,3]. Early weaning can harm their intestinal mucosa, disrupt the natural gut bacteria balance, and raise the likelihood of intestinal infections. Subsequently, piglets may develop inflammation and experience episodes of diarrhea [1,4]. Lipopolysaccharide (LPS), an integral component of Gram-negative bacterial cell walls, is one of the effective stimulators of the immune system [5]. Studies have shown the capacity of LPS to incite the production of pro-inflammatory cytokines, thereby causing an inflammatory response and precipitating the disruption of the intestinal barrier [6]. Therefore, it is feasible to use the LPS immune stress model to establish a model of intestinal barrier injury in piglets. In addition, the gut flora’s composition can affect the gut barrier by influencing the immune system’s development and modulating immune mediators [7].

The aryl hydrocarbon receptor (AhR) is a transcription factor belonging to the basic helix–loop–helix (bHLH) PAS family, and can be activated by natural and synthetic reagents. There is evidence that the AhR plays a pivotal role in regulating gut health [8,9]. 6-formylindolo (3,2-b) carbazole (FICZ) is a ligand that exhibits high affinity for the AhR, effectively protects the intestinal barrier, and relieves intestinal inflammation by inciting AhR activation [10]. FICZ has been demonstrated to improve dextran sulphate sodium (DSS)-induced colitis by reducing IL-6 expression and increasing the expression of tight junction proteins via the activation of the AhR [10,11]. The activation of AhR by FICZ has also been associated with increased Muc2 expression, a higher number of goblet cells, and a reduction in bacterial infiltration, culminating in the mitigation of DSS-induced colitis [12]. AhR activation by FICZ improves intestinal epithelial barrier dysfunction caused by intestinal ischemia/reperfusion (I/R) after hypoxia [13]. Certain natural compounds act as ligands to activate the AhR and play a role in promoting gut health. Cardamonin (CDN), a naturally occurring flavone isolated from Alpinia katsumadai Hayata, exhibits a diverse array of biological activities including anti-inflammatory, anti-bacterial, anti-oxidant, anti-cancer, and anti-viral properties [14,15,16]. A previous inquiry yielded evidence that CDN possesses the ability to function as an AhR ligand, effectively triggering AhR activation to mitigate colonic inflammatory reactions [14].

The objective of this study is to examine the impact of CDN on tight junction proteins, inflammatory response, changes in the intestinal microbiota, and alterations in the metabolomic profile of immune stressed piglets. We hypothesized that CDN may attenuate intestinal barrier damage in immunologically stressed piglets by modulating the intestinal flora and metabolome. This was accomplished by subjecting the piglets to LPS to induce immune stress and employing FICZ as a dependable positive control. The findings herein provide a valuable reference to underpin the nutritional regulation of gut microbiota in weaned piglets, while also offering insights into the prospective utilization of cardamom as a viable feed additive.

## 2. Materials and Methods

### 2.1. Experimental Design, Diet, Sample Collection

Thirty piglets (Duroc × Lancashire × Yorkshire) weaned on day 21 with an average initial body weight of 6.02 ± 0.80 kg were randomly divided into five groups, including the basal diet (Control), basal diet+ lipopolysaccharides (LPS) (*E. coli* serotype 055: B5; Sigma Chemical Inc., St Louis, MO, USA), basal diet + 250 μg/kg FICZ + LPS (FICZ) (Shanghai Yi En Chemical Technology Co., Shanghai China), basal diet + 3 mg/kg CDN (Nanjing Jingzhu Bio-technology Co., Ltd., Nanjing, China) + LPS (LCDN), and basal diet + 6 mg/kg CDN + LPS (HCDN/CDN) groups. The intraperitoneal injection of FICZ or CDN (the injection dose is based on body weight) began on the 8th day of the experiment, with the remaining two groups receiving equal quantities of sterilized saline. Each group consisted of six replicates. The basal diet was formulated following the guidelines from the National Research Council (NRC) in Appendix A. After the 15th day of the initiation of treatment, piglets were subjected to intraperitoneal injection involving 100 μg/kg body weight of LPS or an equivalent volume of sterilized saline. Six hours following the administration of injections, the piglets were humanely euthanized through the administration of sodium pentobarbital solution at 80 mg/kg bodyweight.

The experiment was conducted at the Livestock Research Base of the Chinese Academy of Tropical Agricultural Sciences, and all piglets were provided unrestricted access to both feed and water and were housed in individual enclosures. Piglets were housed in an environmentally controlled room (Temperature range 28 ± 2 °C). The room was cleaned daily and disinfected regularly.

The ileal segments were gently rinsed with normal saline solution. Subsequently, 5 cm samples from the mid-ileum were immersed in 4% paraformaldehyde solution for histomorphological examination. we scraped and collected mid-ileal mucosa samples, froze them immediately in liquid nitrogen, and stored them at −80 °C. Cecal contents were collected using sterile enzyme-free tubes for subsequent intestinal microbiome analysis and metabolomic profiling.

### 2.2. Intestinal Morphology

The fixed ileum specimens were dehydrated in a gradient manner and embedded in paraffin. Following that, the samples that were embedded were divided into sections measuring 5 μm in thickness using a slicer (KD-2258, Jinhua Kedee Instrument Co., Jinhua, China). These sections were subjected to staining using hematoxylin and eosin (H & E) (Boster, Wuhan, China). The final step included sealing the sections with neutral resin for observation under a fluorescence microscope (AxioScope A1, Carl Zeiss, Oberkochen, Germany). For morphological analysis, the measurement of the height of the villi in the intestine (VH) and the depth of the crypts (CD) was conducted using image processing software connected with a light microscope. In each section, 5 distinct fields were chosen randomly for observation and measurement. The VH/CD ratio was calculated to assess the relationship between the height of the intestinal villi and the depth of the crypts.

### 2.3. Immunofluorescence

The paraffin-embedded tissue samples were dewaxed and rinsed thoroughly with distilled water. To enhance the staining intensity of the antibodies, a proteinase K antigen retrieval kit (Boster, Wuhan, China) was used to reveal antigens and epitopes in the fixed tissues. Each sample was then washed using PBS solution at a pH of 7.4, with agitation. The tissue slides were subjected to a blocking process in which they were exposed to a solution containing 5% bovine serum albumin (Solarbio, Bejing, China) for a period of 40 min. Subsequent to this step, a primary antibody (Proteintech, Wuhan, China) was introduced after washing it with PBS, and incubation was extended overnight at 4 °C. After the slides were washed again with PBS, secondary antibodies (FITC-labeled goat anti-rabbit IgG, Cell Signalling Technology, Boston, MA, USA) were added and incubated at room temperature under light-protected conditions. The nuclei were stained with 4′-6-diamidino-2-phenylindole (DAPI, Boster, Wuhan, China) for 10 min at room temperature in the dark. Furthermore, the sample was securely enclosed and treated with a substance that inhibits fluorescence emission (Beyotime Biotechnology, Shanghai, China). The various slices were observed using a fluorescent microscope, and photos were captured for subsequent study.

### 2.4. Real-Time Quantitative PCR Analysis

The extraction of total RNA from the ileum was performed using Trizol reagent (Invitrogen, Carlsbad, CA, USA) in accordance with the manufacturer’s instructions. To validate the concentration and purity of the RNA, we employed a NanoDrop 1000 spectrophotometer and 1% agarose gel electrophoresis. cDNA synthesis was performed in accordance with the methodology outlined in HiScript^®^ III (+gDNA wiper) cDNA Synthesis Kit (Vazyme, Nanjing, China). QuantStudio^TM^ 6 Flex System (Thermo Fisher, Waltham, MA, USA) was utilized to conduct quantitative real-time PCR. The ultimate reaction mixture (10 μL) comprised 5 μL of recently prepared Master Mix, 0.4 μL of the primers, 0.6 μL of cDNA, and 4 μL of diethylpyrocarbonate-treated water. The polymerase chain reaction (PCR) procedure was run in accordance with the manufacturer’s instructions. The 2^−ΔΔCt^ technique, previously reported by Mou et al., was employed to determine the relative mRNA abundance of the investigated genes. For the purpose of normalization, housekeeping genes were selected, with glyceraldehyde-3-phosphate dehydrogenase (GAPDH) being identified as the most consistently expressed reference gene. The baseline value of 1 was assigned to the mRNA levels of each target gene in the control group (CON). The primer sequences employed in this investigation are enumerated in Appendix A.

### 2.5. Western Blotting Analysis

Proteins from the mucosal lining of the ileum were isolated using RIPA lysis buffer (Beyotime Biotechnology, Shanghai, China). The quantification of protein concentration was performed using BCA Protein Assay Kit manufactured by Lagic, a company based in Beijing, China. Following that, the proteins underwent sodium dodecyl sulfate–polyacrylamide gel electrophoresis (SDS-PAGE) and were subsequently deposited onto a polyvinylidene fluoride (PVDF) membrane. In order to facilitate immunodetection, the membrane underwent a blocking process in which it was treated with 5% skimmed milk at room temperature for a duration of 60 min. Subsequently, the membrane was subjected to overnight incubation at a temperature of 4 °C in the presence of primary antibodies. Subsequently, the membrane was subjected to incubation for a duration of 1 h at ambient temperature with the secondary antibody, which was appropriately diluted at a ratio of 1:2000. The immunoreactive protein bands underwent color development using an enhanced chemiluminescence (ECL) kit (NCM Biotech, Suzhou, China), after which the entries were captured using photography. The analysis of the gray levels of the bands was conducted using Image J 2.1 software, developed by the National Institutes of Health in the United States. The primary antibodies utilized in this study were anti-claudin-1 (1:1000), anti-occludin (1:1000), anti-ZO-1 (1:1000), and anti-GAPDH (1:1000) sourced from the Proteintech Group in Wuhan, China. Additionally, goat anti-rabbit IgG (1:2000, Cell Signalling Technology, Boston, MA, USA) was used.

### 2.6. Gut Microbiota Analysis

Genomic DNA was extracted from cecal samples using E.Z.N.A. Stool DNA Kit (Omega Bio-tek, Inc., Norcross, GA, USA). The extraction procedure followed the manufacturer’s instructions. The assessment of DNA quality and concentration was performed using a Nanodrop 2000 spectrophotometer (ThermoFisher Scientific, Inc., Waltham, MA, USA). The V3-V4 regions of the bacterial 16Sr RNA gene were amplified using universal primers (341F: CCTACGGGRBGCASCAG; 806R: GGACTACNNGGGTATCTAAT). To differentiate between distinct samples, 8-base-pair barcode sequences were appended to the 5′ end of both the upstream and downstream primers. These universal primers, bearing their unique barcode sequences, were synthesized and subjected to amplification utilizing an ABI 9700 PCR system (Applied Biosystems, Inc., Waltham, MA, USA). The PCR products were confirmed via 2% agarose gel electrophoresis, followed by automated purification using the Agencourt AMPure XP nucleic acid purification kit (Beckman Coulter, Inc., Brea, CA, USA). Library construction was performed using NEB Next Ultra II DNA Library Prep Kit (New England Biolabs, Inc., Miami, FL, USA). Sequencing was carried out on the Illumina Miseq PE250 platform after the purification and quality inspection of the constructed library. Data were uploaded to the National Center for Biotechnology Information (NCBI) under Bioprojects accession number PRJNA985374.

### 2.7. Short-Chain Fatty Acid (SCFA) Analysis

Two grams of cecal content was added to an EP tube containing two milliliters of water, followed by thorough ultrasonic mixing for 20 min. This was centrifuged at 4 °C for 10 min at 15,000 revolutions per minute (rpm). We collected 1.5 mL of the supernatant and transferred it into a microtube with 0.3 mL of 25% metaphosphoric acid solution. We then centrifuged this mixture at 4 °C for 10 min, at 15,000 rpm. Subsequently, the supernatant was passed through a 0.22 μm filter membrane in order to remove impurities. The resulting filtered liquid was then transferred into a syringe vial for determination. Gas chromatography (7890B, Agilent Technologies, Santa Clara, CA, USA) was utilized to conduct the measurements. 

### 2.8. Ultra-High-Performance Liquid Chromatography Tandem Quadrupole Time-of-Flight Mass Spectrometry (UHPLC-QTOFMS) Analysis 

Briefly, 25 mg of cecal content was added to an EP tube with 500 μL of extract solution. Firstly, the sample was ground for 4 min at 35 Hz and sonicated for 5 min in an ice water bath. This step was repeated three times. Next, we allowed it to stand at −40 °C for an hour, then centrifuged it at 4 °C for 15 min at 12,000 rpm. Finally, we transferred the resulting supernatant into a vial for assaying. The UHPLC procedure was carried out utilizing the Vanquish instrument from Thermo Fisher Scientific. This instrument has the capability to detect a relatively wide range of substances. This system was coupled with a Q Exactive HFX mass spectrometer (Orbitrap MS, Thermo Fisher Scientific, Waltham, MA, USA). Qualitative and quantitative analyses of peaks were carried out using XCMS, self-designed R packages, and proprietary secondary mass spectrometry databases. 

### 2.9. Statistical Analysis

The statistical analysis involved using one-way ANOVA and Duncan’s multiple range test, with SPSS version 26.0 (Chicago, IL, USA). The significance level was set at *p* < 0.05. The findings are reported as mean ± standard error of mean (SEM). To investigate the gut microbiome, the generation of clean tags, subsequent to quality control, splicing, and the removal of chimeric sequences from the raw data, culminated in the clustering of operational taxonomic units (OTUs) using Vsearch (v2.7.1). The alpha diversity index and beta diversity were calculated using QIIME. Significant variations in microorganisms at the phylum and genus levels were determined by employing Wilcoxon tests. Sequence alignment was conducted using BLAST, and representative sequences were annotated against the SILVA138 database. Visualization was accomplished using the R package (v 3.6.0). Regarding metabolomics, differential metabolites were identified based on the following criteria: *p* < 0.05 (student’s *t*-test), a fold change <0.67 or >1.5, and a VIP (Variable Importance in Projection) score exceeding 1.

## 3. Results

### 3.1. Ileal Mucosa Morphology

As shown in Figure 1, LPS was able to alter the intestinal structure of the ileum, while the supplementation of FICZ or CDN demonstrated its ability to reverse this alteration.

To further elucidate these morphological changes in the ileum, we assessed parameters such as crypt depth (CD), villus height (VH), and the ratio of villus height to crypt depth (V/C) (as illustrated in Figure 2). In the LPS-exposed group, both ileal VH and ileal V/C exhibited a significant reduction in comparison with those in the CON group (*p* < 0.05, Figure 2A). Notably, the HCDN group displayed a noteworthy elevation in ileal V/C compared with that of the LPS group (*p* < 0.05, Figure 2C).

### 3.2. Intestinal Barrier Analysis

In comparison to the CON group, the protein and mRNA expression levels of claudin-1, occludin, and ZO-1 were notably reduced in the ileum of the LPS group (*p* < 0.05, Figure 3A and Figure 4A). However, when compared to those of the LPS group, the mRNA and protein expressions of claudin-1 in the ileum were seen to be considerably elevated in the FICZ group (*p* < 0.05, Figure 3C and Figure 4C). Furthermore, the HCDN group demonstrated a significant elevation in the mRNA and protein expressions of ZO-1 and occludin in the ileum (*p* < 0.05, as shown in Figure 3A,B and Figure 4A,B). Moreover, the expression of claudin-1 protein levels shown a significant increase in both the LCDN and HCDN groups (*p* < 0.05). Interestingly, the simultaneous examination of the protein and mRNA expressions of claudin-1 revealed higher levels of these in the FICZ group in comparison with those in both the LCDN and HCDN groups (*p* < 0.05, as depicted in Figure 3C and Figure 4C). The dependability of these data was additionally supported by means of immunofluorescence analysis (as demonstrated in Figure 5).

### 3.3. mRNA Expression of Cytokine

According to the data presented in Figure 6, it is evident that piglets belonging to the LPS group demonstrated a noteworthy elevation in the messenger RNA (mRNA) expression levels of TNF-α and IL-1β (*p* < 0.05), while concurrently displaying a reduction in IL-10 mRNA expression (*p* < 0.05) compared with that in the CON group. In contrast, the administration of FICZ effectively counteracted the LPS-induced upregulation of TNF-α and IL-1β mRNA expression (*p* < 0.05) and prevented a decrease in IL-10 mRNA expression (*p* < 0.05). Both LCDN and HCDN groups exhibited a substantial reduction in interleukin-1β (IL-1β) and an increase in interleukin-10 (IL-10) mRNA expression when compared with those in the lipopolysaccharide (LPS) group (*p* < 0.05). No statistically significant change in IL-6 mRNA expression was observed between the groups (*p* > 0.05).

### 3.4. Gut Microbiota Analysis 

#### 3.4.1. Structure and Diversity of Gut Flora

From the piglet gut microbiota samples, 1,808,705 valid sequences were generated in total. The effective sequences were clustered via UPARSE, which resulted in the generation of operational taxonomic units (OTUs) for the CON, LPS, FICZ, and CDN groups, yielding 2912, 2187, 2953, and 2222 OTUs, respectively (as illustrated in Figure 7A). Alpha diversity analysis showed no significant differences between the groups (Appendix A). Beta diversity analysis demonstrated a significant separation of cecal microbial communities between the groups (Figure 7B).

In the cecum, the dominant bacterial phyla were Firmicutes, Bacteroidota, and Proteobacteria, collectively constituting over 95% of the total cecal bacterial communities (Figure 8A). In comparison with the LPS group, the FICZ and CDN groups had increased Bacteroidota (7.96% vs. 18.11% vs. 17.55%) and decreased Proteobacteria (12.71% vs. 6.82% vs. 10.02%). The LPS group had reduced Bacteroidota (10.61% vs. 7.96%) and increased Proteobacteria (9.40% vs. 12.71%) levels compared with those in the CON group. Further, a detailed examination at the genus level, presented in Figure 8B, shows that in comparison with the control group, the LPS group exhibited an increase in the relative abundance of *Escherichia–Shigella* (3.85% vs. 8.03%) alongside a decrease in *Megasphaera* (0.93% vs. 0.72%). However, intraperitoneal injections of FICZ or CDN reduced the relative abundance of *Escherichia-Shigella* (8.03% vs. 1.60% vs. 3.16%) and increased the amount of *Megasphaera* (0.72% vs. 2.38% vs. 2.66%). 

To identify statistically significant intestinal bacteria, the Wilcoxon test was performed. At the genus level, *Bacteroides*, *Prevotella*, *Prevotellaceae_NK3B31_group*, and *Succinivibrio* were significantly upregulated in the FICZ group compared with those in the LPS group (*p* < 0.05, Figure 9A). *Bacteroides*, *Megamonas*, and *Succinivibrio* showed a notable increase in the CDN group (*p* < 0.05, Figure 9B).

#### 3.4.2. PICRUSt2-Predicted Functional Pathway Analysis

To evaluate the effect of intraperitoneal injections of FICZ and CDN on the intestinal bacterial community of LPS-stressed piglets, we employed the PICRUSt2 method to investigate the KEGG pathway composition of the intestinal microbiota. Figure 10A shows that intraperitoneal injection of FICZ significantly increased a relative abundance of genes such as nicotinate and nicotinamide metabolism, protein digestion and absorption, other gylcans degradation, biosynthesis of vancomycin group antibiotics, N-Glycan biosynthesis, zeatin biosynthesis, and sphingolipid metabolism (*p* < 0.05). Similarly, intraperitoneal injection of CDN also significantly increased niacin and nicotinamide metabolism, sphingolipid metabolism, and zeatin biosynthesis, but reduced the relative abundance of genes for unsaturated fatty acid gene biosynthesis (*p* < 0.05, Figure 10B).

### 3.5. Short-Chain Fatty Acid (SCFA) Analysis

To further validate these findings, the assessment of short-chain fatty acids was conducted through gas chromatography (Table 1). The level of iso-butyric acid in the LPS group decreased (*p* < 0.05) compared with that in the CON group. However, there was no significant change in the content of other short-chain fatty acids. The FICZ group had higher levels of acetic acid, propionic acid, butyric acid, iso-butyric acid, and iso-valeric acid compared with the LPS group. The contents of acetic acid, propionic acid, and butyric acid in the HCDN group, and the propionic acid contents in the LCDN group were also increased in comparison with those in the LPS group (*p* < 0.05).

### 3.6. Metabolomics Analyses

To identify the intergroup differential compounds, the parameters of *p* value < 0.05, VIP ≥ 1, and fold change < 0.67 or >1.5 were used as criteria. Compared to the LPS group, the FICZ group exhibited significant increases in 29 metabolites and significant decreases in 38 metabolites (as delineated in Figure 11A). Similarly, in the CDN group, significant increases were observed in 9 metabolites, while 73 metabolites showed significant decreases (Figure 11B).

Upon thorough screening, the FICZ group had increased levels of metabolites such as anethole, uridine 5-diphosphogalactose, quercetin, betaine, saikosaponin b2, pachymic acid, and dihydrocapsaicin. Conversely, it exhibited reduced levels of 15-ketoiloprost and 15-cyclohexylpentanorprostaglandin f2alpha in comparison with those in the LPS group (Table 2). The CDN group displayed increased levels of metabolites like tetrahydrocorticosterone, lappaconitine, hexanoyl-l-carnitine, crotonic acid, and telmisartan while having reduced levels of “13,14-dihydro-15-ketotetranorprostaglandin e2”, “adenosine 3,5-cyclic monophosphate”, thromboxane b2, “11-deoxy-16,16-dimethylprostaglandin e2”, and prostaglandin b1 (Table 3).

### 3.7. Correlation Analysis

As shown in Figure 12, the relative abundance of *Bacteroides* was observed to be positively correlated with uridine 5′-diphosphogalactose, anethole, saikosaponin b2, and indolelactic acid. *Prevotella* was positively correlated with acetic acid, propionic acid, and butyric acid, and negatively correlated with 15-cyclohexylpentanorprostaglandin f2alpha. *Prevotellaceae_NK3B31_group* was positively correlated with betaine, and negatively correlated with 15-cyclohexylpentanorprostaglandin f2alpha. *Succinivibrio*, *Cloacibacillus*, *Erysipelotrichaceae_UCG_009*, and *Sutterella* were positively correlated with acetic acid and propionic acid, and negatively correlated with 15-ketoiloprost (Figure 12A). *Megamonas* had a positive correlation with crotonic acid and tetrahydrocorticosterone, but had a negative correlation with 11-deoxy-16,16-dimethylprostaglandin e2 and “13,14-dihydro-15-ketotetranorprostaglandin e2”. *Succinivibrio* were positively correlated with butyric acid, crotonic acid, and tetrahydrocorticosterone. *Succinivibrio* were negatively correlated with “adenosine 3,5-cyclic monophosphate” and “13,14-dihydro-15-ketotetranorprostaglandin e2” (Figure 12B).

## 4. Discussion

### 4.1. AhR Activation Improves Ileal Mucosa Morphology in Immune-Stressed Piglets

The morphology of the intestines plays a vital role as a mechanical barrier, and is indicative of the general health and absorptive capability of the intestinal system. Changes in the structure of the intestines, such as the shrinking of villi and an increase in the size of crypts, are suggestive of impaired nutrient absorption and inhibited growth [17,18]. Decreases in villus height (VH) have been identified as an indicator of intestinal toxicity caused by LPS [17]. Furthermore, the intestinal morphology and health of piglets can also be assessed using the Velvet VH:CD ratio [6]. In the present investigation, the administration of LPS resulted in the development of acute intestinal damage, as evidenced by a decrease in the villous height (VH) and villous-to-crypt (V/C) ratios, specifically in the ileum of the LPS-exposed piglets.

Remarkably, supplementation with either CDN or FICZ effectively countered these adverse effects on ileal morphology induced by LPS. This observation is consistent with previous studies that demonstrated that dietary supplementation with the AhR ligand indole-3-methanol (I3C) in mice alleviated the abnormal intestinal mucosal morphology associated with AhR deficiency. Moreover, it has been discovered that the induction of the aryl hydrocarbon receptor (AhR) pathway through the use of indole-3-acetic acid (IAA) has the ability to regulate the equilibrium of the intestinal environment and enhance the immune response inside the mucosal lining of mice [19,20]. Thus AhR activation positively affected ileal mucosal morphology in immunotressed piglets, mitigating the negative impacts of inflammation on the integrity of the intestinal tract. Notably, our experiment revealed that the administration of 6 mg/kg CDN had a more pronounced beneficial impact on ileal mucosal morphology compared with other treatments.

### 4.2. AhR Activation Reduces Intestinal Barrier Damage in Immune-Stressed Piglets

The intestinal barrier comprises a layer of the columnar epithelium and tight junctions that connect the individual epithelial cells. Tight junctions serve as the primary cellular constituents responsible for preserving tissue integrity and barrier functionality. The composition of this structure is characterized by its intricate molecular makeup, which encompasses transmembrane protein complexes, including claudins and occludins, as well as cytosolic proteins like ZO (junctional adhesion molecule ZO-1, ZO-2, and ZO-3). The aforementioned components constitute a structural border located at the intersection of neighboring cells. They function as rate-limiting factors in the paracellular pathway and serve as a permeable barrier [21]. These proteins are essential for epithelial barrier function [22]. In the context of inflammatory bowel disease, it is observed that there is a significant reduction in the expression of tight junction proteins. This reduction leads to an increased permeability of the intestinal lining to harmful compounds, which in turn triggers a systemic inflammatory response [23]. Hence, it is imperative to sustain the integrity of the intestinal epithelial barrier in order to prevent the onset of intestinal ailments and inflammatory conditions. Previous research has demonstrated that FICZ effectively counteracted the decrease in the expression levels of ZO-1, occludin, and claudin-1 induced by DSS in mice colitis, thereby enhancing the integrity of the intestinal barrier [11]. The activation of the aryl hydrocarbon receptor (AhR) in the mouse intestine resulted in a considerable rise in the expression levels of ZO-1, occludin, and claudin-4 [24]. Furthermore, the stimulation of the AhR by compounds derived from indole resulted in a significant improvement in the functionality of the epithelial barrier in the intestines. The observed enhancement was ascribed to an increase in ZO-1 and claudin-1 gene expression [25]. In the current study, we observed significant upregulation in the mRNA and protein expression of ZO-1, occludin, and claudin-1 in the ileum of immunostressed piglets following the intraperitoneal injection of FICZ or CDN. The upregulation described in this context functions to augment the integrity of the intestinal barrier, illustrating the protective effects of activating the aryl hydrocarbon receptor (AhR) in the maintenance of intestinal health.

### 4.3. AhR Activation Reduces mRNA Expression of Cytokine in Immune-Stressed Piglets

Numerous studies have shown that cytokines play an important role in the process of immune stress in piglets. Significantly elevated levels of pro-inflammatory cytokines, such as IL-1β, IL-6, and TNF-α, are an important feature of inflammation in the body, whereas the anti-inflammatory cytokine IL-10 counteracts immune cell activation and attenuates the inflammatory response [26]. Following intestinal AhR activation, it can modulate intestinal immune responses by promoting the production of transforming growth factor β (TGF-β) and IL-10 by intestinal epithelial cells, while inhibiting the production of inflammatory factors TNF-α, interleukin-22 (IL-22), interleukin-35 (IL-35), and interferon-gamma (IFN-γ) by immune cells, which can affect the function of intestinal epithelial and immune cell function [27,28]. In vitro assays on BV2 microglia have shown that Cardamonin is a potential inflammation inhibitor with immunomodulatory effects on the production of pro-inflammatory cytokines (TNF-α, IL-6, PGE2, IL-1β, and NO) [29]. Another study has found that Cardamonin treatment attenuated intestinal disorders, including recurrent colitis and colitis-associated tumors, and reduced IL-1β and TNF-α secretion [15]. The results of this experiment showed that in the ileum, the intraperitoneal injection of FICZ significantly decreased the mRNA expression of TNF-α and IL-1β and increased the mRNA expression of IL-10 in immunostressed piglets, whereas the intraperitoneal injection of CDN in immunostressed piglets increased the content of IL-10 and decreased the content of IL-6. The experimental results indicated that AhR activation could alleviate the inflammation level of the ileum and enhance the anti-inflammatory capacity of immunostressed piglets. Overall, the intraperitoneal injection of 3 mg/kg of CDN was more effective in enhancing the anti-inflammatory ability of ileal mucosa in immunostressed piglets.

### 4.4. AhR Activation Improved Intestinal Microbial Diversity and Short-Chain Fatty Acid Content in Piglets under Immune Stress

Intestinal flora and its metabolites have a regulatory role in host physiology. In this trial, LPS induced changes in the gut microbiota composition without significantly affecting the α diversity index. This aligns with the results of Xiao et al. [30] and Geng et al. [31], but differs from the results of Wen et al. [32]. These differences could be attributed to variations in nutritional conditions and individual differences. In line with previous discoveries, Firmicutes, Bacteroides, and Proteobacteria remain the dominant phyla in the pig gut [30,33,34]. Research has shown that a low abundance of *Bacteroides* and its associated metabolic dysfunction can be considered microbial biomarkers for post-weaning physiological diarrhea in piglets [35]. Both FICZ and CDN significantly increased *Bacteroides*, a major short-chain fatty acid (SCFA) producer. SCFAs are recognized for their significant contribution to the regulation of gut health. Both acetate and propionate are potent anti-inflammatory mediators [36]. Butyrate serves as the main source of energy for colon cells, and a lack of butyrate can cause harm to the intestinal barrier [37]. *Prevotella* is an important dietary fiber fermenter in the pig digestion tract and it belongs to the phylum Bacteroides [38]. *Succinivibrio* is an SCFA-producing bacterium whose relative abundance is significantly reduced in non-surviving hemodialysis patients [39]. Studies have shown that butyrate and FICZ synergistically activate the AhR [40]. In this study, compared with those in the LPS group, SCFA-producing microorganisms such as *Bacteroides*, *Prevotella*, *Prevotellaceae_NK3B31_group*, and *Succinivibrio* in the FICZ group were enriched and accompanied by increased levels of SCFAs. The CDN group exhibited an increase in microorganisms that produce SCFAs, specifically *Bacteroides*, *Megamonas*, and *Succinivibrio*, accompanied by increased levels of acetic acid, propionic acid, and butyric acid. The microbial enrichment of SCFAs in the FICZ group and CDN group was consistent with the change in SCFA levels. This aligns with the results of Modoux et al. [40]. Predictive results from PICRUSt2 indicated that the functional abundance of pathways including nicotinate and nicotinamide metabolism, sphingolipid metabolism, and zeatin biosynthesis were significantly increased in the FICZ and CDN groups compared with those in the LPS group. The control of niacin and nicotinamide metabolism plays a crucial role in combating inflammation [41]. Sphingolipids, a class of lipids with vital roles in membrane biology and intestinal inflammation, have received much attention [42]. Moreover, zeatin nucleosides have been scientifically proven to effectively inhibit the synthesis of pro-inflammatory cytokines in the immune cells of mice [43].

### 4.5. AhR Activation Regulates the Metabolome in Immune-Stressed Piglets

The effect of AhR ligands on the intestinal metabolic profile was investigated using UHPLC-QTOFMS in this work. Quercetin, saikosaponin b2, and dihydrocapsaicin exhibited significant enrichments in the FICZ group. Quercetin is a ligand with a high affinity for the human AhR, and can mitigate DSS-induced intestinal barrier damage and significantly reduce inflammation [44,45]. Both saikosaponin b2 and dihydrocapsaicin effectively suppress the activation of inflammation-related pathways induced by LPS [46,47]. Tetrahydrocorticosterone, lappaconitine derivatives, and telmisartan exhibited significant enrichments in the CDN group. Tetrahydrocorticosterone is a corticosteroid associated with the maintenance of the water–electrolyte balance [48]. Both lappaconitine derivatives and telmisartan have anti-inflammatory effects [49,50]. Increases in anti-inflammatory substances may be associated with a reduction in intestinal inflammation and thus intestinal barrier damage. The synthesis and accumulation of prostaglandins and their derivatives are closely linked to the inflammatory response. In this study, FICZ treatment reduced the production of 15-ketoiloprost and 15-cyclohexylpentanorprostaglandin f2alpha. CDN treatment reduced inflammatory prostate substances and their derivatives such as “13,14-dihydro-15-ketotetranorprostaglandin e2”, “11-deoxy-16,16-dimethylprostaglandin e2”, and prostaglandin b1. Shawn [51] et al. demonstrated that certain prostaglandins can serve as natural ligands for the AhR. High prostaglandin levels were found to trigger AhR transformation and DNA binding in laboratory settings. However, this activation may lead to the stimulation of additional AhR pathways, subsequently inducing inflammation [52]. Correlation analyses are useful in assessing the degree of interconnectedness between gut flora and various metabolites. Positive correlations were found between *Succinivibrio* and several SCFAs, while negative correlations were found between the former and prostaglandin-related derivatives. *Bacteroides* was observed to be positively correlated with uridine 5′-diphosphogalactose, anethole, saikosaponin b2, and indolelactic acid. This suggests that FICZ and CDN could reduce intestinal barrier damage in immunostressed piglets by modulating the intestinal flora and metabolic profile. Specific mechanisms require further study.

## 5. Conclusions

Our research results show that CDN administration can effectively improve intestinal mucosal barrier damage and the inflammatory response caused by LPS. A dose of 6 mg/kg of CDN is better. In addition, we believe that CDN may alleviate LPS-induced intestinal barrier damage by regulating the composition and function of the intestinal microbiota. This finding provides new clues for further exploring the potential of CDN in the treatment of intestinal inflammation.

## Figures and Tables

**Figure 1 animals-14-00794-f001:**
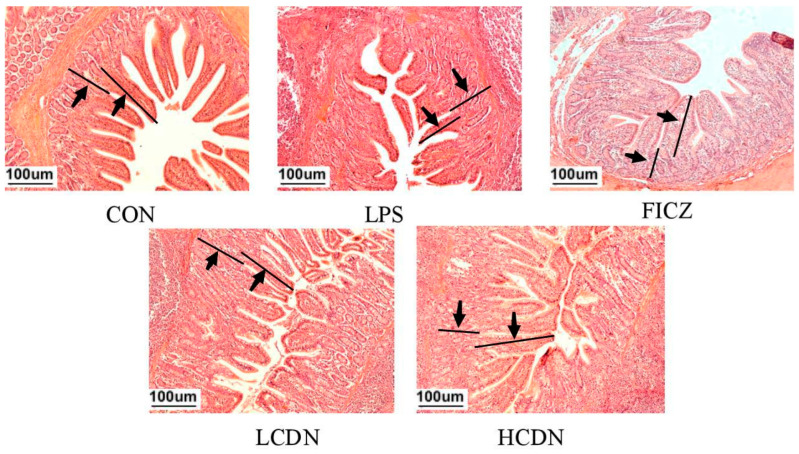
Ileum histomorphological observations. CON: control group; LPS: pre-slaughter injection of LPS; FICZ: intraperitoneal injection of 250 μg/kg FICZ and pre-slaughter injection of LPS; LCDN and HCDN: intraperitoneal injection of 3 mg/kg CDN or 6 mg/kg CDN and pre-slaughter injection of LPS. Arrows mark the observation area.

**Figure 2 animals-14-00794-f002:**
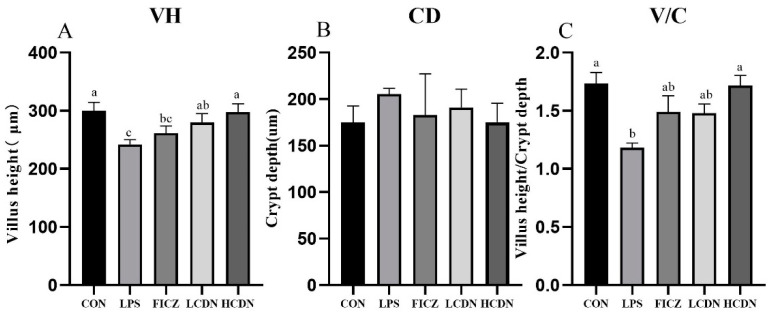
Morphological indicators of the ileal mucosa. CON: control group; LPS: pre-slaughter injection of LPS; FICZ: intraperitoneal injection of 250 μg/kg FICZ and pre-slaughter injection of LPS; LCDN and HCDN: intraperitoneal injection of 3 mg/kg CDN or 6 mg/kg CDN and pre-slaughter injection of LPS. Significant difference between groups not containing the same letter.

**Figure 3 animals-14-00794-f003:**
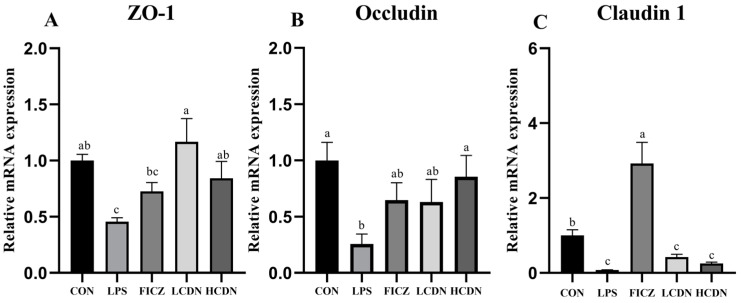
The relative mRNA expression of tight junction proteins in the ileal mucosa of piglets. CON: control group; LPS: pre-slaughter injection of LPS; FICZ: intraperitoneal injection of 250 μg/kg FICZ and pre-slaughter injection of LPS; LCDN and HCDN: intraperitoneal injection of 3 mg/kg CDN or 6 mg/kg CDN and pre-slaughter injection of LPS. Significant difference between groups not containing the same letter.

**Figure 4 animals-14-00794-f004:**
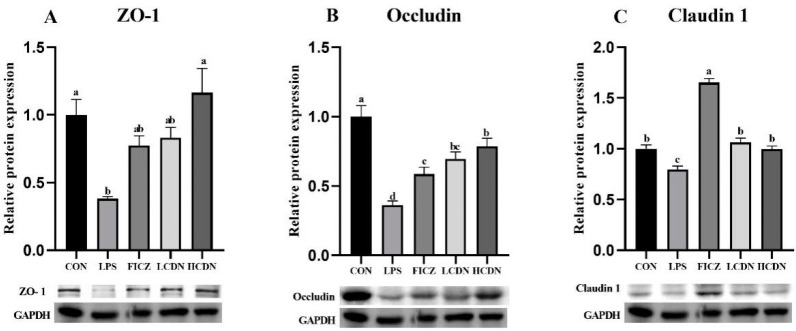
Expression of tight junction proteins in the ileal mucosa of piglets. CON: control group; LPS: pre-slaughter injection of LPS; FICZ: intraperitoneal injection of 250 μg/kg FICZ and pre-slaughter injection of LPS; LCDN and HCDN: intraperitoneal injection of 3 mg/kg CDN or 6 mg/kg CDN and pre-slaughter injection of LPS. Significant difference between groups not containing the same letter.

**Figure 5 animals-14-00794-f005:**
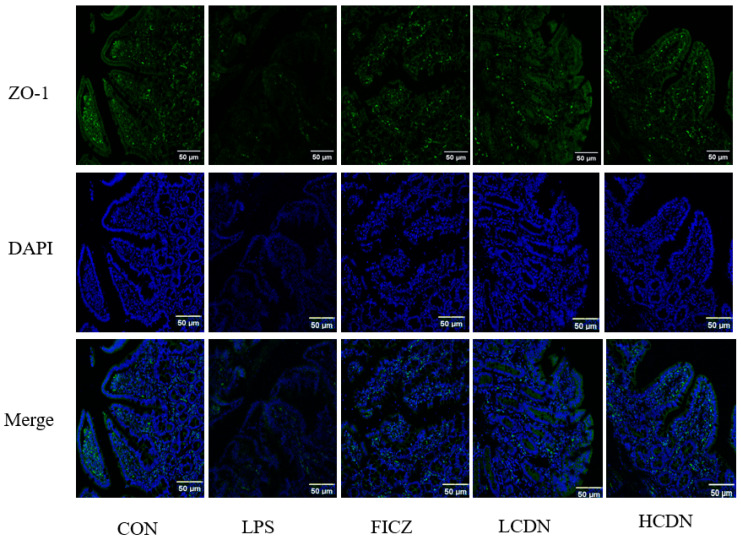
Ileal immunofluorescence staining of piglets. CON: control group; LPS: pre-slaughter injection of LPS; FICZ: intraperitoneal injection of 250 μg/kg FICZ and pre-slaughter injection of LPS; LCDN and HCDN: intraperitoneal injection of 3 mg/kg CDN or 6 mg/kg CDN and pre-slaughter injection of LPS.

**Figure 6 animals-14-00794-f006:**
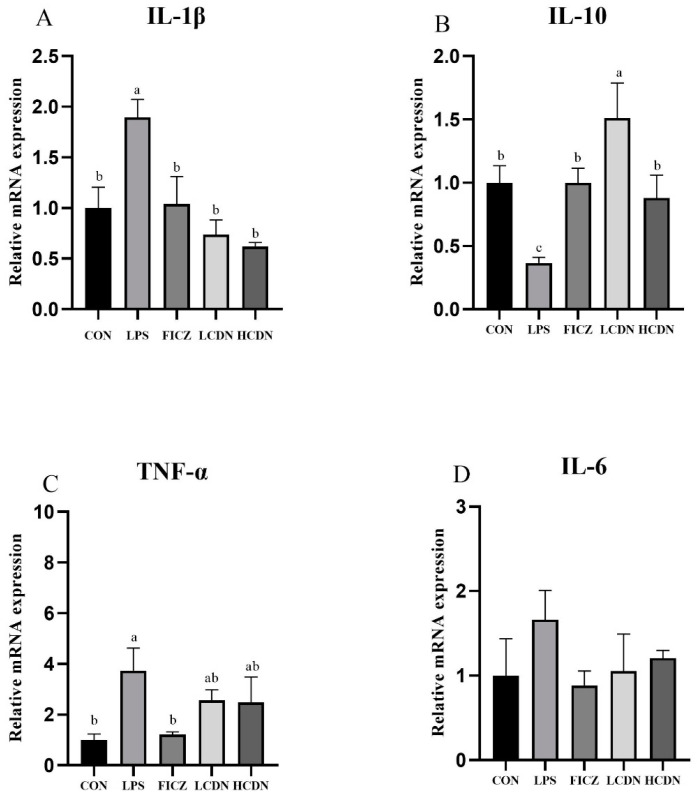
The relative mRNA expression of cytokines in the ileal mucosa of piglets. CON: control group; LPS: pre-slaughter injection of LPS; FICZ: intraperitoneal injection of 250 μg/kg FICZ and pre-slaughter injection of LPS; LCDN and HCDN: intraperitoneal injection of 3 mg/kg CDN or 6 mg/kg CDN and pre-slaughter injection of LPS. Significant difference between groups not containing the same letter.

**Figure 7 animals-14-00794-f007:**
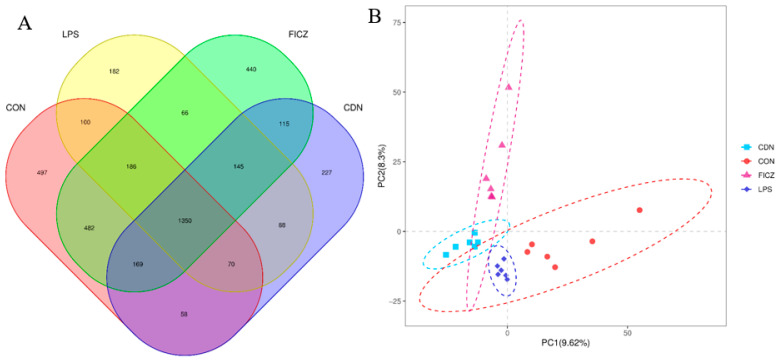
Analysis of microbial operational taxonomic units (OTUs) and beta diversity. (**A**) Venn diagram of operational taxonomic units (OTUs); (**B**) partial least squares discriminant analysis (PLS-DA) of beta diversity. Each color represents a group (*n* = 6).

**Figure 8 animals-14-00794-f008:**
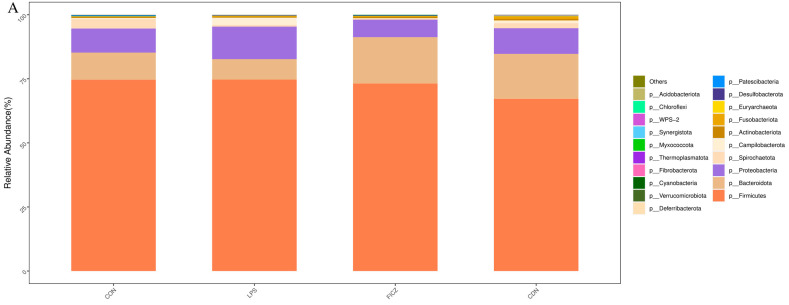
The phylum (**A**) and genus (**B**) levels of cecal flora that were analyzed.

**Figure 9 animals-14-00794-f009:**
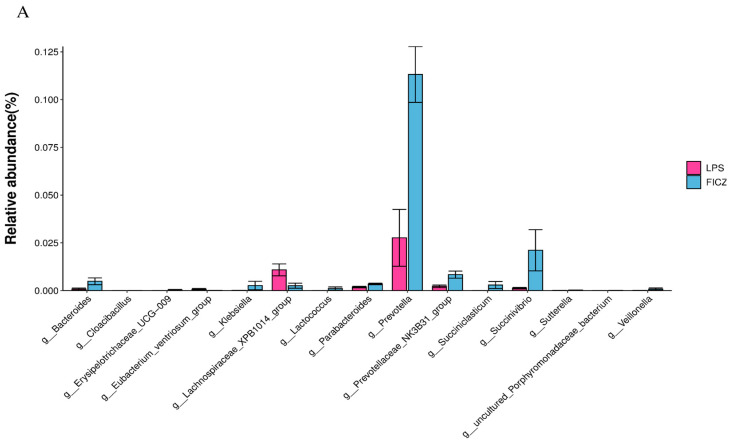
Differential bacterial genera based on the Wilcoxon test. LPS vs. FICZ (**A**); LPS vs. CDN (**B**). CON: control group; LPS: pre-slaughter injection of LPS; FICZ: intraperitoneal injection of 250 μg/kg FICZ and pre-slaughter injection of LPS; CDN: intraperitoneal injection of 6 mg/kg CDN and pre-slaughter injection of LPS.

**Figure 10 animals-14-00794-f010:**
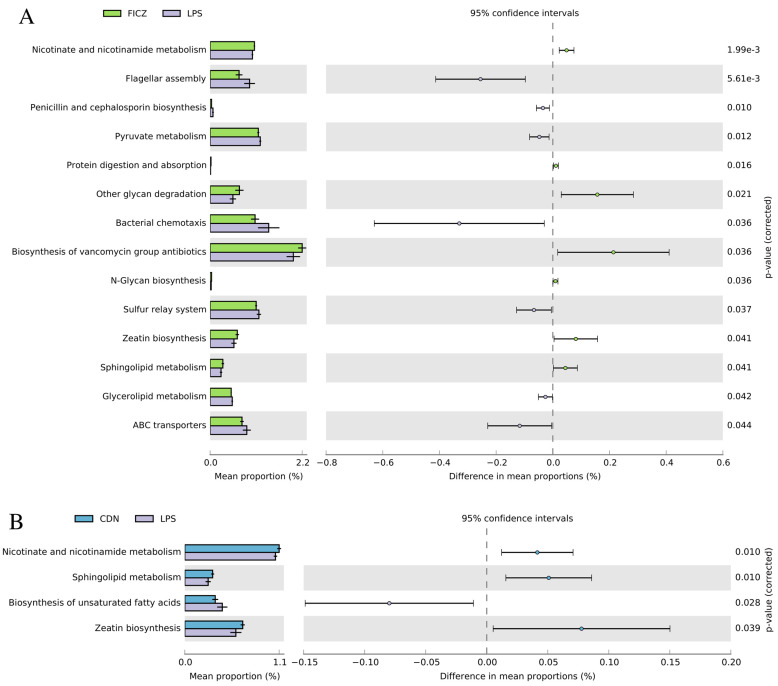
Gut microbiome comparison in PICRUSt2-predicted functional pathway analysis; the third level of the KEGG pathway is shown in the extended error bar. The *p*-values are shown on the right. “KEGG level” refers to the hierarchical structure of the KEGG pathway. The KEGG pathway is divided into different levels, usually from Level 1 to Level 3. Different levels provide different levels of detail. Level 3 of the pathway provides more detailed information.

**Figure 11 animals-14-00794-f011:**
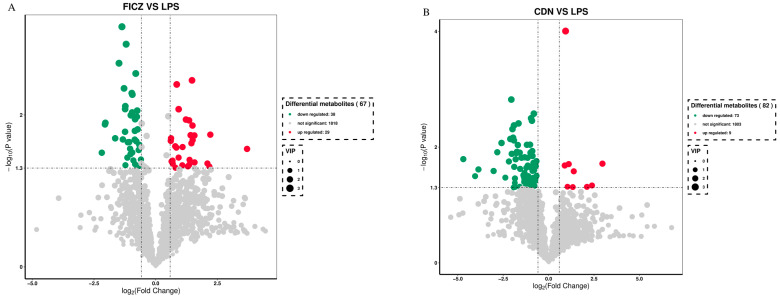
Volcano plot of differential metabolites of weaned piglets in the FICZ (**A**) and CDN (**B**) treatment groups. CON: control group; LPS: pre-slaughter injection of LPS; FICZ: intraperitoneal injection of 250 μg/kg FICZ and pre-slaughter injection of LPS; CDN: intraperitoneal injection of 6 mg/kg CDN and pre-slaughter injection of LPS.

**Figure 12 animals-14-00794-f012:**
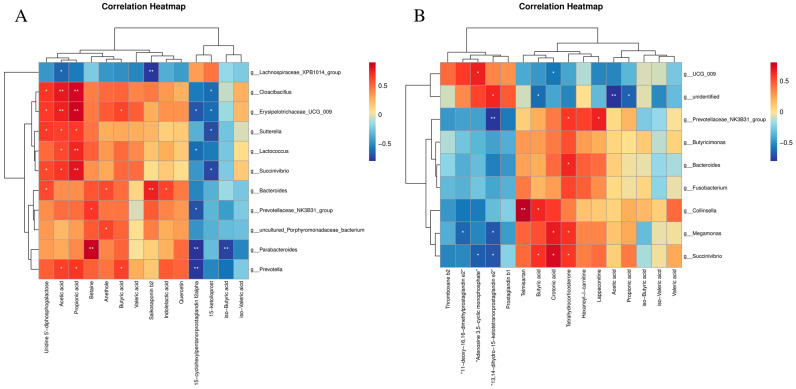
(**A**) Correlation analysis between gut microbiota and metabolomics. LPS-FICZ; (**B**) LPS-CDN. * *p*-value < 0.05; ** *p*-value < 0.01.

**Table 1 animals-14-00794-t001:** Content of short-chain fatty acids (SCFAs).

Items mg/g		Groups			
	CON	LPS	FICZ	LCDN	HCDN
Acetic acid	3.26 ± 0.16 ^bc^	2.59 ± 0.11 ^c^	7.73 ± 0.15 ^a^	4.35 ± 0.23 ^bc^	4.98 ± 0.11 ^b^
Propionic acid	2.24 ± 0.15 ^bc^	1.74 ± 0.07 ^c^	5.96 ± 0.09 ^a^	2.86 ± 0.25 ^b^	2.95 ± 0.16 ^b^
iso-Butyric acid	1.31 ± 0.06 ^a^	0.15 ± 0.02 ^b^	0.13 ± 0.02 ^b^	0.20 ± 0.03 ^b^	0.19 ± 0.07 ^b^
Butyric acid	1.26 ± 0.03 ^bc^	0.83 ± 0.04 ^c^	2.84 ± 0.04 ^a^	1.44 ± 0.10 ^bc^	1.78 ± 0.04 ^b^
iso-Valeric aicd	0.14 ± 0.002 ^b^	0.33 ± 0.06 ^a^	0.29 ± 0.02 ^a^	0.42 ± 0.04 ^a^	0.33 ± 0.01 ^a^
Valeric acid	0.37 ± 0.03 ^b^	0.28 ± 0.03 ^b^	1.32 ± 0.04 ^a^	0.43 ± 0.07 ^b^	0.41 ± 0.06 ^b^

Each value represents the mean ± SEM. Means values with different letters within a row indicate a significant difference (*p* < 0.05).

**Table 2 animals-14-00794-t002:** Identification of metabolite statistics; differential metabolites in the FICZ vs. LPS group.

MS2 Metabolite	VIP	*p*-Value	Regulated	LOG-FOLDCHANGE
Dihydrocapsaicin	2.0112	0.021934	up	0.61289
Confertifoline	1.8944	0.040915	up	0.64403
“6,7,4-trihydroxyisoflavone”	1.8556	0.039896	up	0.68289
Chaulmoogric acid	1.9898	0.025845	up	0.79199
Oleic acid methyl ester	1.8356	0.049591	up	0.80598
“D-lysergic acid n,n-diethylamide”	1.9823	0.026908	up	0.83793
Pachymic acid	2.2856	0.0083937	up	0.93002
Cholestenone	1.9460	0.026444	up	1.0893
Arg-Ile	1.8466	0.047509	up	1.3036
Saikosaponin b2	2.1335	0.011783	up	1.355
Nicotinate d-ribonucleotide	1.8537	0.038902	up	1.3711
Betaine	1.9961	0.023602	up	1.4485
D-ribose 1-phosphate	2.1018	0.013782	up	1.4909
Quercetin	2.1244	0.018647	up	1.5692
2-(n-ethyl-n-m-toluidino)ethanol	1.8608	0.043479	up	2.0975
Uridine 5-diphosphogalactose	1.8457	0.047818	up	2.1773
D-fructose 6-phosphate	2.0381	0.018179	up	2.2142
Anethole	2.0317	0.028003	up	3.7029
4-imidazoleacrylic acid	2.5018	0.002083	down	−1.4865
Phenol	2.6445	0.00069024	down	−1.3677
15-ketoiloprost	2.2770	0.010094	down	−1.0058
15-cyclohexylpentanorprostaglandin f2alpha	2.3686	0.0051695	down	−0.96624
4-imidazoleacrylic acid	2.5018	0.002083	down	−1.4865

**Table 3 animals-14-00794-t003:** Identification of metabolite statistics; differential metabolites in the CDN vs. LPS group.

MS2 Metabolite	VIP	*p*-Value	Regulated	LOG-FOLDCHANGE
Pantetheine	2.0731	0.0161	down	−4.6970
“13,14-dihydro-15-ketotetranorprostaglandin e2”	1.8806	0.0317	down	−4.0492
Dihydrofolic acid	2.0272	0.0177	down	−2.1853
“Adenosine 3,5-cyclic monophosphate”	2.3800	0.0072	down	−2.1068
“2,7,8-trimethyl-2-(beta-carboxyethyl)-6-hydroxychroman”	2.4622	0.0015	down	−2.0551
Thromboxane b2	1.8902	0.0349	down	−1.1719
Dapsone	1.8302	0.0459	down	−1.1707
“11-deoxy-16,16-dimethylprostaglandin e2”	2.2233	0.0091	down	−1.0219
1-palmitoyl-2-lauroyl-sn-glycero-3-phosphorylcholine	2.4830	0.0026	down	−0.8113
Prostaglandin b1	1.9438	0.0212	down	−0.7051
Tetrahydrocorticosterone	2.0868	0.0207	up	0.8899
“1-(5-fluoropentyl)-1h-indazole-3-carboxylic acid, 1-naphthalenyl ester”	2.8076	0.0001	up	0.9329
Asn-Trp-Arg	2.0571	0.0196	up	1.1073
Glu-Asn-Arg	1.8234	0.0491	up	1.3299
Lappaconitine	1.9794	0.0262	up	1.3863
Hexanoyl-l-carnitine	1.8998	0.0486	up	2.1091
Crotonic acid	1.8925	0.0460	up	2.3880
Telmisartan	2.0992	0.0193	up	2.9631

## Data Availability

The data that validate the findings of this study can be freely accessed from the National Center for Biotechnology Information (NCBI), with the BioProject accession number PRJNA985374.

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
