# Peer review of "AhR Activation Ameliorates Intestinal Barrier Damage in Immunostressed Piglets by Regulating Intestinal Flora and Its Metabolism"

_animals, 2024, doi:10.3390/ani14050794_

Round 1

Reviewer 1 Report

Comments and Suggestions for Authors

The manuscript sent for review covers a very important field.  However, the authors have not been protected from many errors. Comments on the manuscript are given below. I believe that the manuscript needs to be revised and go through the review process again.

1)      CRITICAL NOTE: The introduction are short and the authors get straight to the details, lacking an introduction to topics such as pig breeding, economic importance etc.

2)      CRITICAL NOTE: The composition of compound diets should be in the manuscript not as one table. They should be prepared for all mixtures. This is an essential element in this type of research. Furthermore, the authors should specify the environmental conditions in detail and not in general terms. This entire subsection should be rewritten.

3)      CRITICAL NOTE: Materials and methods - There is no explanation as to why fragments were not taken from the duodenum and jejunum. This is very surprising as it is mainly these intestinal fragments that are used in this type of experiment. Especially as the authors discuss nutrient absorption at the beginning of the discussion.

4)      CRITICAL NOTE: The subsection "2.2. Intestinal Morphology" was described in a very laconic way. It is not specified on what (on what equipment) these 5um sections were prepared. Was this a microtome? If so, which one. The authors should have specified in detail what measurements they took and at what place. They also specified the VH/CD ratio - on what basis did they determine it - was it taken from literature data? If so the authors should add citations. Furthermore, HE staining is an inadequate staining to measure the parameters specified in the manuscript (it does not give the correct contrast), hence this raises questions about the accuracy of their performance. The authors should include higher magnification microscopic images in the manuscript so that they can verify whether the measurements could have been properly assessed.

5)      CRITICAL NOTE: The authors selectively indicate the manufacturers of the reagents (e.g. for DAPI it is given - DAPI, BOSTER, AR1176) and already for e.g. bovine serum albumin no such information is given. This applies to the entire MM section. The authors should revise the description of the Materials and Methods section and add the manufacturers for each reagent in the text or add a reagents subsection at the end of the Materials and Methods section and list all manufacturers there.

6)      CRITICAL NOTE: Why was only the cecum used for some analyses (e.g. Short-chain fatty acids (SCFAs) analysis or Ultra High Performance Liquid Tandem Chromatography Quadrupole Time of Flight Mass 184 Spectrometry UHPLC-QTOFMSanalysis)? Why not also the ileum?

7)      CRITICAL NOTE: To use one-way analysis of variance, the conditions of normal distribution and homogeneity of variance must be present. The authors should specify which tests were used for this analysis.

8)      CRITICAL NOTE: Results - figure 1 - microscope images should be shown under higher magnification. In addition, the manner in which the measurements were taken should be indicated on the microphotographs (e.g. with arrows, length marking).

9)      NOTE: Results - figure 5 - scale not marked on microphotographs

10)   NOTE:  In the graphs, the difference markings should be in a larger size font, as the image needs to be enlarged to see them.

11)   CRITICAL NOTE: The discussion lacks reference in relation to its own research findings. The chapter should be rewritten to include the research results obtained and compare them with literature data.

Reviewer 2 Report

Comments and Suggestions for Authors

Cardamonin ameliorate intestinal barrier damage in immune stressed piglets by regulating intestinal flora and its metabolism. ANIMALS

 Very interesting work, many analytical techniques were used to obtain reliable and objective results from many sides. The issue is important for practice because the problem observed during piglets weaning (loss of appetite, indigestion, slow growth, diarrhea, death) remains unsolved and brings large economical losses to breeders. Manuscript needs the minor revision.

-Line 22: correlation, instead of Correlation.

-L 27: avoid using abbreviation in key words (AhR; FICZ)

Material and methods:

- the experimental design is not clear. Please, provide each experimental action using the day of piglet’s age (starting from the 21st day of age, when the experiment began). The use of "days of age" and "days of experiment" is confusing. (For example: 21st day of age – start of experimental diet, 29th day of age - intraperitoneal injection of FICZ or CDN, …., Xth day of age – six hours following the administration of injections, the piglets were humanely euthanized through the administration of sodium pentobarbital solution at 80 mg/kg body weight. The experiment lasted ….. days.)

- L 73-86: what does it mean “…/kg”? kg of feed or kg of body weight? The information should be added through the whole chapter.

- Please, indicate on what day of age the piglets were euthanized.

- What was the sex of the piglets and how were the sexes divided in the groups?

- Table S1: how did you calculate the digestible energy? Explain the „Puffed soybean”.

- L 90: from which segment of ileum the „samples of ileal mucosa” were collected? From mid-ileum? It should be added.

Results:

I think the subtitles (for ex.: AhR activation improves ileal mucosa morphology in immune-stressed piglets; AhR activation reduces intestinal barrier damage in immune-stressed piglets) are much more suitable for Discussion. For the Results chapter I suggest for example „Ileal mucosa morphology; Intestinal barier analysis; mRNA expression of cytokine; and so on.

- Fig. 7A: the numbers inside the figure are not visible.

Round 2

Reviewer 1 Report

Comments and Suggestions for Authors

I stand by my comments that I made in an earlier version of the manuscript. From my point of view, they are comments that should be included in the text so that it meets the requirements of a scientific paper. Unfortunately, the authors have only improved some of them. In my opinion, the paper is very interesting, but it needs more improvement on the part of the authors.
